# Low-Rank Adaptation with Swin Transformers enhance Skin Cancer Diagnosis

**Prasanth Yadla**
**Independent Researcher**
pyadla2@alumni.ncsu.edu

**Editors:** Accepted for publication at MIDL 2025

## Abstract

Skin cancer is one of the most common forms of cancer worldwide. Automated diagnosis using deep learning has shown promise, but high-performing models like Vision Transformers are often computationally expensive. Swin Transformers are less computationally expensive than ViTs because they use a hierarchical structure with shifted windows for self-attention, limiting computations to local regions instead of the entire image. We propose a Parameter Efficient Fine-tuning (PEFT) method integrating Low-Rank Adaptation (LoRA) into Swin Transformers to reduce model training and inference computational complexity while maintaining high diagnostic performance. Experiments on the standard HAM10000 skin cancer dataset demonstrate the proposed model's effectiveness in skin lesion classification with improved efficiency.

**Keywords:** Skin cancer, Swin Transformer, LoRA, Vision Transformer, Medical image classification, HAM10000

## 1. Introduction

Skin cancer, encompassing both melanoma and non-melanoma types, represents a significant global health concern, with early and accurate diagnosis being crucial for effective treatment. Traditional visual diagnosis methods, however, are often hindered by subjectivity and inter-observer variability. Over the years, deep learning techniques have greatly improved diagnostic performance for medical image classification, with models like ResNet (He et al., 2015) and DenseNet (Huang et al., 2017) playing a central role in skin lesion analysis. However, these models often struggle with capturing long-range dependencies in images, limiting their ability to fully model the complex spatial relationships present in medical images.

To address this limitation, Vision Transformers (ViTs), and specifically Swin Transformers, have emerged as a promising alternative. Swin Transformers leverage shifted window attention mechanisms and hierarchical structures, which allow them to capture global contextual information more effectively than traditional architectures. However, the computational demands of Swin Transformers are significant, posing challenges for their practical use in large-scale applications, even though their attention mechanism is linear in time complexity.

To overcome this challenge, we propose integrating Low-Rank Adaptation (LoRA) into Swin Transformers for the task of skin cancer classification. LoRA reduces the number of trainable parameters by incorporating low-rank matrices into the attention weights, enabling efficient fine-tuning without sacrificing performance.

Recent advancements in transformer-based models, such as SkinSwinViT (Tang et al., 2024), have demonstrated the potential of Swin Transformers in dermoscopic image classification. Other transformer-based architectures, including MedT (Valanarasu et al., 2021) and UNETR (Hatamizadeh et al., 2021), have been explored for segmentation and classification tasks in medical imaging.

In addition, parameter-efficient methods like Adapter-BERT (Houlsby et al., 2019), Bit-Fit (Zaken et al., 2021), and LoRA (Hu et al., 2021) have been developed to enable scalable model adaptation with minimal computational cost. While LoRA has been less explored in medical imaging, it holds significant promise for fine-tuning large models efficiently, making it a powerful solution for skin cancer classification tasks.

## 2. Methods

We adopt the Swin Transformer as the backbone for our classification model. Swin Transformer operates with shifted windows and hierarchical feature maps, making it well-suited for visual recognition tasks.

LoRA is applied to the self-attention layers of the Swin Transformer. Given a weight matrix $W \in \mathbb{R}^{d \times k}$, LoRA approximates it as:

$$W' = W + AB,$$

where $A \in \mathbb{R}^{d \times r}$ and $B \in \mathbb{R}^{r \times k}$, and $r \ll \min(d, k)$. Only $A$ and $B$ are trainable during fine-tuning, while $W$ is frozen.

### 2.1. Dataset

We train and evaluate our model using the HAM10000 ("Human Against Machine with 10000 training images") dataset, which is a benchmark dermoscopic image dataset for skin lesion analysis. The dataset consists of 10,015 high-resolution dermoscopic images collected from different populations and stored in a consistent format. These images are annotated with seven classes of skin lesions: Melanocytic nevi (nv), Melanoma (mel), Benign keratosis-like lesions (bkl), Basal cell carcinoma (bcc), Actinic keratoses (akiec), Vascular lesions (vasc), and Dermatofibroma (df).

The dataset is balanced through stratified sampling to mitigate class imbalance during training. Each image is preprocessed by resizing to 224×224 pixels and normalized using ImageNet statistics. We randomly split the dataset into 80% training, 10% validation, and 10% testing sets while ensuring a balanced distribution across lesion types.

### 2.2. Training Details and Hyperparameters

We used the base Swin Transformer model from Hugging Face, implemented in PyTorch 2.4, and trained (fine-tune) it on a single NVIDIA RTX 3090 GPU (24GB VRAM) using mixed precision (FP16) and Unsloth kernels for faster computation and reduced memory usage. The base Swin Transformer (Swin-B) has approximately 87.7 million parameters and was pretrained on ImageNet-1K. It uses a patch size of 4×4, an embedding dimension of 96, and a window size of 7. The model architecture includes 4 stages with depths [2, 2, 18, 2], a multilayer perceptron (MLP) ratio of 4, GELU activations, and LayerNorm for

normalization. This configuration offers a balance of efficiency and performance, making it well-suited for vision tasks like skin cancer classification. Training with HAM10000 dataset was conducted over 50 epochs with early stopping (patience of 10 epochs) based on validation F1-score. The Adam optimizer was used with a learning rate of $1 \times 10^{-4}$, a batch size of 32, and a weight decay of $1 \times 10^{-5}$. A dropout rate of 0.1 was applied to reduce overfitting. The learning rate followed a cosine annealing schedule with 500 warm-up steps. LoRA was used for parameter-efficient fine-tuning, and we experimented with LoRA ranks 4, 8, and 16. The LoRA scaling factor was set to 32. LoRA was applied to query and value target matrics within the attention kernels of all the transformer blocks. The hyperparameters were selected based on grid search and prior work in transformer-based medical imaging tasks. In addition, data augmentation techniques included random flipping, cropping, rotation (up to 20°), color jittering, and normalization using ImageNet statistics.

## 3. Results

Performance is assessed using accuracy, precision, recall, and F1-score.

Table 1 presents the comparison between baseline Swin Transformer and the proposed LoRA-enhanced version.

Table 1: Classification Performance on HAM10000 Dataset with Varying LoRA Ranks

| Model | Accuracy | Precision | Recall | F1-score |
|---|---|---|---|---|
| Swin Transformer | 85.7% | 84.3% | 83.9% | 84.1% |
| Swin + LoRA ($r = 4$) | **87.4%** | **86.1%** | **85.6%** | **85.8%** |
| Swin + LoRA ($r = 8$) | 87.1% | 85.8% | 85.3% | 85.5% |
| Swin + LoRA ($r = 16$) | 86.6% | 85.2% | 84.7% | 84.9% |

The LoRA-integrated model achieves improved accuracy with a significantly reduced number of trainable parameters on a single GPU with FP32 precision. (Table 2).

Table 2: Model Complexity Comparison

| Model | Trainable Parameters | Training Time (per epoch) |
|---|---|---|
| Swin Transformer (Swin Base) | 88M | 4.2 min |
| Swin + LoRA (r=4) | 28M | 2.5 min |

## 4. Conclusion

This work presents an efficient transformer-based model for skin cancer diagnosis by integrating Low-Rank Adaptation into Swin Transformers. Our model maintains high classification performance while significantly reducing the training burden, making it suitable for clinical settings with limited computational resources. Future work includes expanding to other medical image tasks and exploring hardware-specific optimizations.

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
