# OpenReview forum: "Low-Rank Adaptation with Swin Transformers to Enhance Skin Cancer Diagnosis"
_MIDL.io/2025/Short_Papers — MIDL 2025 - Short Papers_

### Official Review · Reviewer_ben1 · 2025-04-28

**Rating:** 4
**Confidence:** 4

**Summary:**

The paper shows that fine-tuning only small Low-Rank Adaptation (LoRA) matrices inside a Swin-Base Transformer can cut trainable parameters by roughly two-thirds and shorten each training epoch, yet still raise accuracy by about two percentage points on the HAM10000 dermoscopy dataset. The idea is practical and clearly described, but the experiments are limited to one dataset and one baseline, so the overall contribution feels incremental.

**Strengths:**

+ Efficiency win without accuracy loss – meets its stated goal: similar (slightly better) performance at much lower computational cost.
+ Addresses a real bottleneck: limited GPU memory and compute in many medical settings.
+ Method is easy to reproduce; hyper-parameters, ranks, and hardware are fully listed.
+ Reports both performance and resource usage, giving a balanced view of trade-offs.
+ Brings a popular NLP technique (LoRA) into the medical-imaging domain.

**Weaknesses:**

1. Single-dataset evidence – only HAM10000 is used; no cross-dataset test (e.g., ISIC 2018) to show generalisability.
2. Baseline breadth – compares only to a full-fine-tuned Swin. Efficient CNNs (ResNet-50, EfficientNet-B4) or lightweight ViTs (LeViT) are absent, so readers can’t judge the true efficiency frontier.
3. Clinical metrics missing – overall accuracy can hide class imbalance; melanoma-specific recall and AUROC at high sensitivity are not reported.
4. No LoRA ablation – adapters are placed in every layer; an experiment on partial insertion or different ranks vs. performance would clarify trade-offs.

---

### Decision · Program_Chairs · 2025-05-01

Accept